# Design of a Dual-Purpose Patch Antenna for Magnetic Resonance Imaging and Induced RF Heating for Small Animal Hyperthermia

**Donghyuk Kim** [1], **Daniel Hernandez** [2] and **Kyoung-Nam Kim** [1,2,*]

1    Department of Health Sciences and Technology, GAIHST, Gachon University, Incheon 21999, Korea; qnrrmrtjdeye@gmail.com
2    Department of Biomedical Engineering, Medical Campus, Gachon University, Incheon 21936, Korea; theasdwmove@gmail.com
*    Correspondence: kyoungnam.kim@gachon.ac.kr; Tel.: +82-32-820-4263; Fax: +82-32-460-8230

**Abstract:** The popularity of patch antennas in magnetic resonance imaging (MRI) has reduced because of the large size required for patch antennae to resonate. Since the size of the patch antenna is associated with the wavelength and the wavelengths that are used in MRI are substantially large, large antennas are used. Methods of reducing patch antenna sizes have been proposed; however, these methods reduce the penetration depth and uniformity. In this study, we reduced the area of the patch antenna by 30% by folding the ground and patch planes in a zigzag pattern. The patch antenna produced two main resonant modes. The first mode produced a uniform magnetic field that was used for MRI. The second mode produced a strong and focused electric ($|E|$)-field, which was used for radiofrequency (RF) heating. Furthermore, we explored the use of a combination of two patch antennas aligned along the $z$-axis to provide a circular uniform magnetic flux density ($|B_1|$) field at 300 MHz, which corresponds to the Larmor frequency in the 7T MRI system. In addition, the patch antenna configuration will be used for RF heating hyperthermia operating at 1.06 GHz. The target object was a small rat with insertion of colon cancer. Using the proposed configuration, we achieved $|B_1|$-field uniformity with a standard deviation of 3% and a temperature increment of 1 °C in the mimic cancer tissue.

**Keywords:** MRI; patch antenna RF heating hyperthermia; full body MRI

## 1. Introduction

Preclinical animal models play an important role in understanding of human disease. It is applied to a variety of research, ranging from diagnosis methods and treatments for diseases. Especially Magnetic resonance imaging (MRI) is a medical imaging method that allows the acquisition of anatomical images from small animals [1–3]. MR images can be employed to diagnose cancer and tumor tissues [4–7]. In addition, MRI is the predominant method for performing non-invasive temperature measurement during hyperthermia treatments [8–12]. Scientists study cancer properties and treatments using ultra-high field MRI, and preclinical studies in controlled environments are usually performed with small animals such as rats and mice [12,13]. In order to provide a stable hyperthermia protocol in clinical practice, it is necessary to accurately measure changes in energy transfer parameters (applied power level, heating duration) in small animal experiments, for which we integrate a system of heating and temperature monitoring. Most heating experiments of small animal integrated with MRI have been implemented in clinical MR scanners. Since the animal MRI system has a smaller bore size than the clinical MRI system, there is a technical limitation in integrating it with a heating device in the limited space [14].

The results of previous works [15,16] indicate the usefulness of heat treatment induced into tumor cells of small animals through a heating system. In previous studies, focused

ultrasound, radiofrequency (RF) and microwave (MW) were used as heating systems that can be combined with MRI systems [17–21]. Specifically, RF or MW energy is applied to the target area in the form of an electric ($|E|$)-field through various types of antennas such as slots, dipoles, bow ties, etc. [18–22] to induce a temperature rise of ~43 °C or higher in the target tissue. In combination with radiotherapy and/or chemotherapy the hyperthermia treatments with an increased exposure time has shown to produce constant cell death rate [22]. In ref. [19] it is shown that the use of a Yagi-Uda antenna for hyperthermia applications operating at 433 MHz, it exhibited low coupling with the imaging RF coils operating at 64 MHz. This is an important aspect when designing an RF transmitter for hyperthermia. Similar to Yagi-Uda antennas, Patch antennas are a type of electromagnetic field generator that consist of a dielectric material placed between the ground and patch plane. At frequencies above 400 Mhz, there are also studies for the purpose of thermal treatment of patch antennas [23–25]. Even though these studies were combined with an MRI system, MR images acquisition and heating were operated separately. MR image acquisition and heating using the SAR value for a bow-tie antenna that can operate in two ways has been proposed [21].

In this study, we present a dual-purpose patch antenna that can be used for MRI imaging and RF heating hyperthermia. To do that, we need to solve a few problems. The first problem is that at 300 MHz, the 7T MRI operating frequency, the size of the antenna is too large to be applied to small animals. The second issue is the $|B_1|$-field profile according to the orientation of the antenna, as the $|B_1|$-field consists of the $B_x$ and $B_y$ components of the magnetic field. For MRI, RF coils that can transmit and receive a uniform and strong $|B_1|$-field are preferred. If the antenna is placed on the $z$–$y$ plane, as is common for any RF coil, the $|B_1|$-profile has a low intensity because the major component of the magnetic field in this orientation is the z component of the magnetic field $B_z$, which is not part of the $|B_1|$-field. To solve these problems, we first reduced the size of the patch antenna such that it is suitable for applications with small animals, such as a rat; second, we aimed to combine the reduced patch antenna in a configuration to improve the imaging $|B_1|$-field and simultaneously produce a strong focused electric field that can be used for RF heating. The proposed antenna configuration has the advantage that it has the potential to be used in small animal MRI bore and in human size scanner; this is possible since the size of the antenna is small, and it can perform imaging and RF heating with the same device. We evaluated the temperature increase using the proposed antenna.

## 2. Methods

### 2.1. Patch Antenna without Size Reduction

Patch antennas consist of a patch and a ground plane made of a conductor material, in between the patch and the ground plane a dielectric material is placed. The patch plane is typically rectangular, and the resonant mode of the patch antenna depends on the wavelength. The width and length of the patch plane and the height of the dielectric material for a specific frequency are calculated using the formula for patch antenna found in [23]. For the case of a patch antenna operating at a frequency of 300 MHz, as used in a 7T MRI system, and if the dielectric material has a permittivity $\varepsilon_r$ = 6.45 and height of 30 mm, the computed values for the width would be 258.9 mm and length 209.9 mm; such an antenna is show in Figure 1a. The resonant modes after matching to 50 Ohms of the patch antenna are shown in Figure 1b. The first resonant frequency is at 300 MHz, and the second is located at 430 MHz. The magnetic $|B_1|$-field pattern of this antenna at 300 MHz in the $y$–$z$ plane is show in Figure 1c, and the corresponding electric field at 429 MHz is shown in Figure 1d.

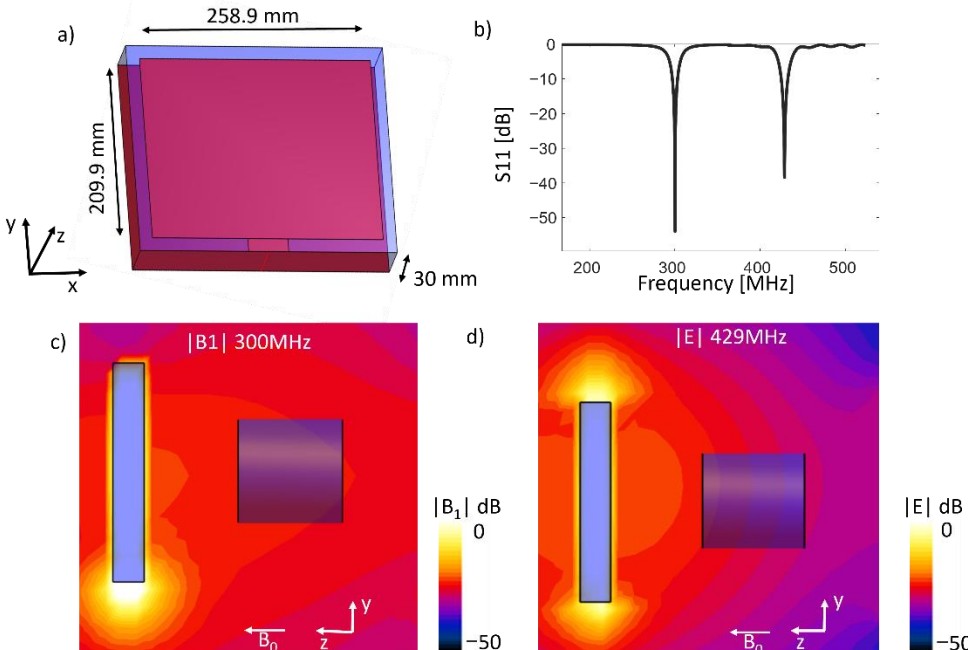

**Figure 1.** Design of (**a**) a patch antenna without size reduction, (**b**) $S_{11}$ parameters of the antenna showing dual frequency, (**c**) the $|B_1|$-field of the antenna at 300 MHz and (**d**) the $|E|$-field at 429 MHz.

This dual resonance frequency can be exploited such that the first resonant frequency can be used for transmitting and receiving in the MRI procedure and the second frequency can be employed for hyperthermia applications.

### 2.2. Patch Antenna with Size Reduction

Size reduction methods for patch antennas have been actively researched [26–30]. One of the most common methods to reduce the size of the antennas is to use meander patterns [30]. As described in the introduction, the resonant modes of the patch antenna are related to the frequency and size of the antenna. Thus, we propose reducing the size of the antenna using the effective length and width of the antenna [24,25,31], by folding the antenna in a zigzag manner. The width of the antenna is reduced by meandering the ground plane in a number of steps (*n*) in the horizontal direction so that the plane consists of two horizontal and two vertical lines. The height of each step is *hx*, and the length of each step is *lx* such that the effective length can be described by the following equation:

$$W = 2n(hx + lx) \tag{1}$$

Figure 2 shows the application of this equation and the meandering method. By choosing *n* = 5 and *hx* = 10 mm, *lx* becomes 18.76 mm, which makes the total width of the antenna 187.6 mm. A similar equation and method can be used to reduce the length of the antenna, and in this work, we applied meandering on the patch plane in the vertical direction. A non-uniform meander patter will allow us to design the antenna to have a straight patch plane at the center, as to uninterrupted conductor area closer to the phantom or the rat so that the field can be of higher intensity. To reduce the length of the antenna, we opted to have a non-uniform meander pattern on the patch plane.

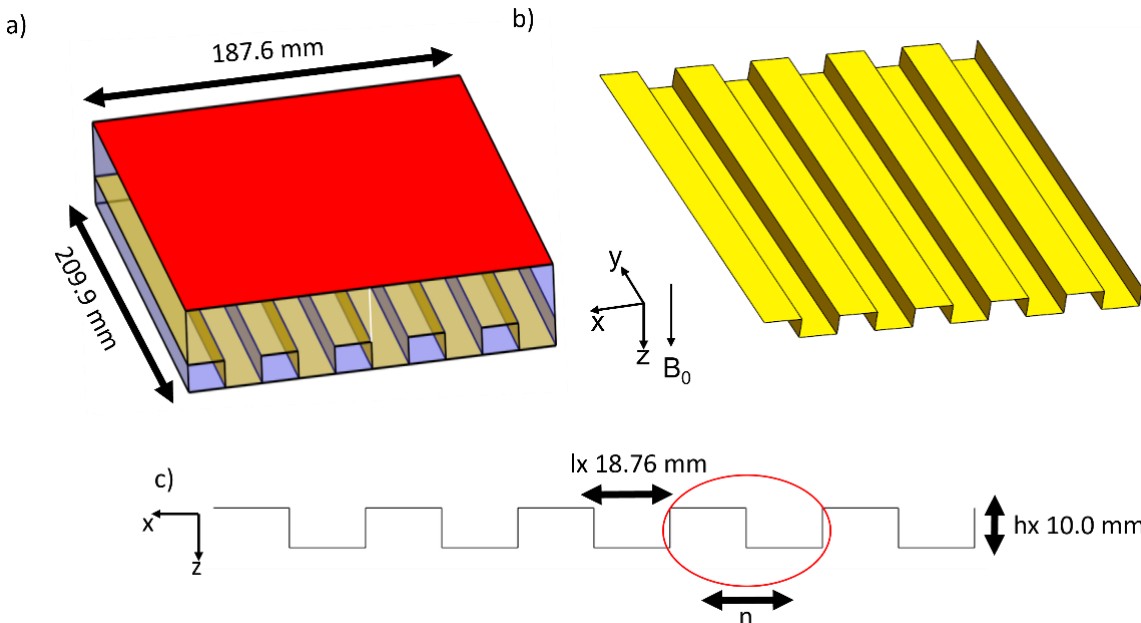

**Figure 2.** The design of the patch antenna (**a**) with reduction in the horizontal direction, (**b**) the ground plate showing the meandering pattern and (**c**) the diagram of the meandering pattern and dimensions.

We further reduced the size of the antenna by applying a number of steps $n = 10$ in the horizontal direction and $n = 3$ in the vertical direction, for the ground and patch plane, respectively. The size of the horizontal step $hx$ was 10 mm and $lx$ was 4.5 mm, the vertical $hy$ was set to 15 mm and $ly$ was 32.5, and 20 mm for the small steps and 50 mm for the larger plane in the middle of the antenna. Figure 3 shows the final geometry. The height, width and length of the dielectric material were 30, 99 and 150 mm, respectively. In Figure 3b the ground plane is shown with the corresponding meandering pattern, and Figure 3c for the top plane of the patch.

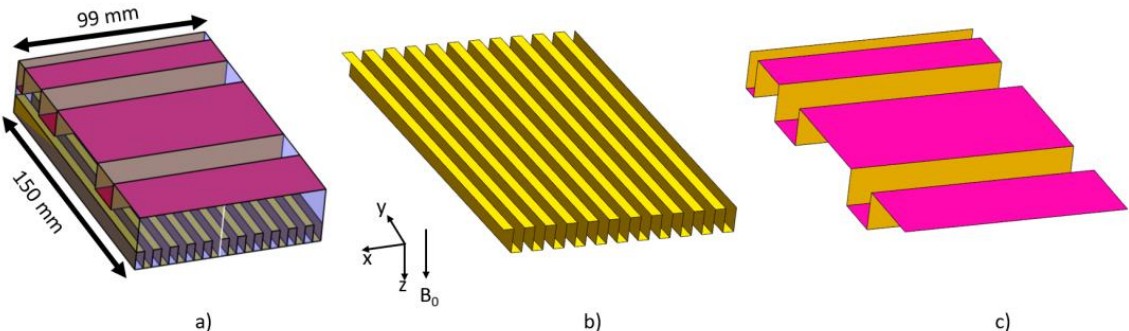

**Figure 3.** (**a**) The reduced patch antenna in the vertical and horizontal directions, (**b**) the ground plane meandering pattern and (**c**) the patch (**top**) plane meandering pattern.

### 2.3. Simulation Setup

We performed EM simulations using the commercial software Sim4Life (Zurich med tech, https://zmt.swiss/sim4life/, accessed on 10 March 2021). Sim4Life is an FDTD based software to solve Maxwell's equations and for electromagnetic analysis, it also provides computation of the temperature based on the RF excitation from an antenna. Two types of EM simulations were performed, one for acquiring a $|B_1|$-field operating at 300 MHz for MRI applications and the other to produce an electric field at 1.06 GHz which was consequently used to compute the temperature rise in the target cancer-mimicking tissue. We used the antennas shown in Figure 4, and the dielectric material selected was a Rogers

RT duroid 6006 with a conductivity of 0.0015 S/m and permittivity of 6.45 at 300 MHz and 1.06 GHz. We excited the antennas with a Gaussian pulse with a central frequency of 300 MHz and 600 MHz bandwidth, and the second simulation using a central frequency of 1.06 GHz with a bandwidth of 2 GHz.

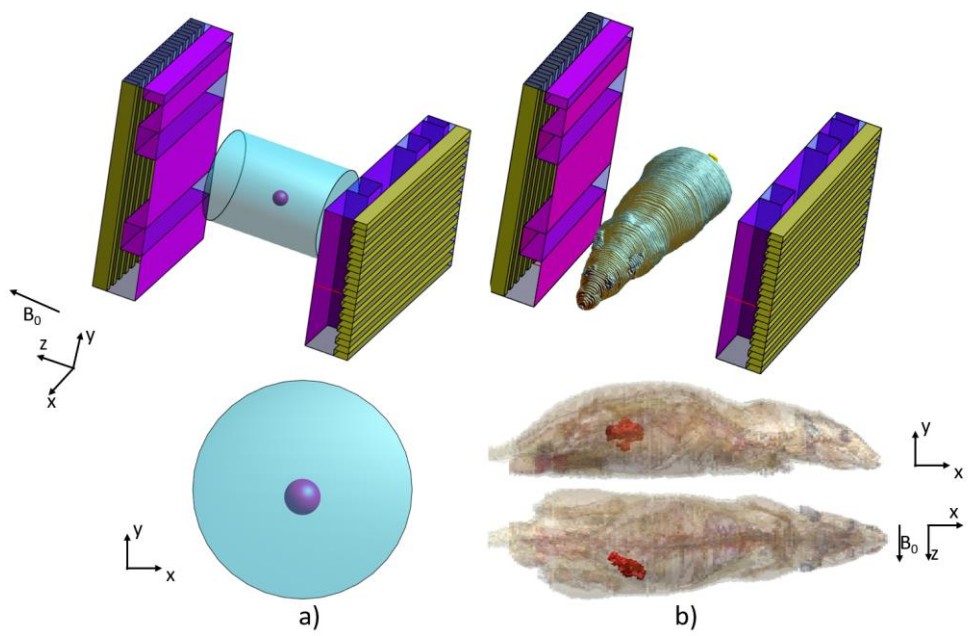

**Figure 4.** The arrangement of the double patch antenna (**a**) for a cylindrical phantom and (**b**) a rat model, showing the cancer-mimicking tissue.

To produce a uniform magnetic field, we used two patch antennas positioned along the *z*-axis and exited with a phase difference such that the field between them was highly uniform. For this purpose, we placed a second patch antenna and rotated it by 90° phase with around the *z*-axis. By having two antennas in the *x*–*y* plane and applying a 90° phase difference, a uniform circular polarized magnetic field can be produced.

Using this configuration, a circular magnetic field was produced in the middle of the antennas. With this configuration, a focused |E|-field can be achieved. Figure 4 shows the arrangement of the double patch antennas. In Figure 4a the cylindrical phantom is shown with the patch antennas. Figure 4b shows the rat model with the orientation between the patch antennas, and also indicating in highlighted red color the cancer-mimicking voxels.

A cylindrical phantom with a radius of 30 mm and a length of 80 mm was positioned along the *z*-axis to demonstrate the capabilities of the patch antenna arrangement, with electrical properties equivalent to the tissues of the large intestine with an electrical conductivity and permittivity of 0.77 S/m and 58.20, respectively, at 300 MHz. The spherical cancer-mimicking tissue is a sphere of 5 mm radius that was placed in the middle of the cylinder phantom. The cancer-mimicking tissue had a conductivity of 1.05 S/m and permittivity of 62.85, these values were based on previously reported results on electrical properties measurements of rodent's cancers [32]. For RF heating, the electrical properties were updated using the database and model provided by the Sim4life. The values set at 1.06 GHz for the cylindrical phantom were 1.0 S/m and 54.7 for electrical conductivity and permittivity, respectively. To update the values of the spherical cancer-mimicking tissue we used the ref. [32] for which the conductivity was set to 1.2 S/m and the permittivity to 59.6. Furthermore, we used the computed |E|-field to estimate the temperature change inside the phantom. The changes in the induced temperature (*T*) of a tissue over time (t) are described by the Pennes bioheat equation (PBE) [21,33], which is given by

$$\rho C_p \frac{\partial T}{\partial t} = \nabla \cdot (k \nabla T) + Q_m + SAR + \rho_b C_{p,b} \rho \omega_b (T_a - T) \tag{2}$$

where $k$ is the thermal conductivity in W/(m·K), T is the tissue temperature in °C, $Q_m$ is the specific metabolic heat generation rate, SAR is the specific absorption rate, $C_p$ is the heat capacity of the tissues in J/(kg·K) and $\omega_b$ is the perfusion rate given in 1/s. $\rho$ is the tissue density (kg/m$^3$); $\rho_b$, $C_{p,b}$ and $T_a$ are the density, specific heat capacity and the temperature of the blood, respectively. All these values are specific to each tissue. The specific absorption rate SAR is related to the energy in the form of the |E|-field deposited into the tissue and is calculated by the electric field:

$$SAR = \frac{\sigma \cdot |E|^2}{\rho} \tag{3}$$

where $\sigma$ is the electrical conductivity (S/m), and $\rho$ is the tissue density (kg/m$^3$). As the equations indicate, the temperature depends on the electric field and the conductivity of the tissue. Therefore, a strong and focused |E|-field is desired. For this simulation, we used the Pennes algorithm for 200 s, and the thermal parameters for the tissues were assigned automatically using the Sim4Life software database. A power of 5 W was applied enough to increase the temperature by 1 °C.

Furthermore, we applied the patch antenna configuration to a rat of 196 mm × 50 mm × 55 mm with a mass of 198 g. Rat 3D models were provided by the animal model from the library of Vizoo (ITIS foundation, Zurich, https://itis.swiss/, accessed on 18 February 201), by using the "Big Male Rat". This model consisted of 52 tissues with the respective electrical properties. The boundary condition at the rat were set with the background of Neumann boundary condition with a Heat Flux of 0 W/m$^2$, and the initial condition for the air was set to 25 °C. The use of the Neumann boundary condition fixes the heat flux at the interface; the sign of this value determines whether the flux flows into or out of the material. This boundary condition used with a heat flux of 0 W/m$^2$ results in thermal insulation of the boundary, removing the need for water bolus. The skin had a heat transfer rate of 7969.16 W/m$^3$/K. In the rat model, we also included a cancer-mimicking tissue, which was the target tissue for the RF heating system, the cancer tissue had an electrical conductivity and permittivity of 1.05 S/m and 62.85, respectively, at 300 MHz and electrical conductivity and permittivity of 1.2 S/m and 59.6 at 1.06 GHz, the conductivity and permittivity values of the cancer tissue was selected based on the measurements in ref. [32]. The cancer tissue was modeled by using part of the large intestine, and the final model was carried out by intersecting a sphere and the large intestine, separating the normal tissue and the cancer tissue by changing the electrical properties as described previously. We used the same simulation setup as the phantom simulations for both the EM and temperature computations, except for using a power of 10 W.

## 3. Results

By using the method of folding the ground and patch plane, it was possible to reduce the size of the antenna to a size of 99 by 150 mm. Figure 5 shows the $S_{11}$ parameters of the proposed antenna configuration; despite the size reduction, the two-resonance mode was still present.

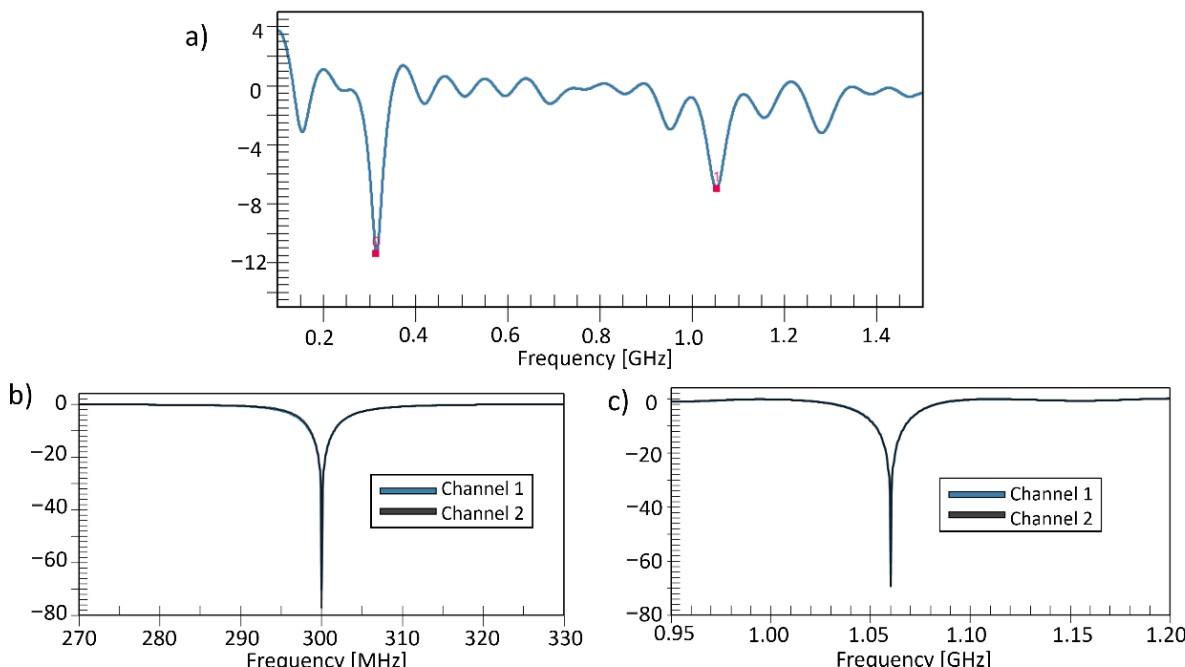

**Figure 5.** The S$_{11}$ parameters of the proposed patch antenna, (**a**) with two main resonance frequencies of 300 MHz and 1.06 GHz. performed 50 ohm matched (**b**) at 300 MHz and (**c**) at 1.06 GHz each of two proposed patch antennas.

We performed 50 ohms matching of the two channels to apply for the MRI application and RF heating application. Separate matching circuits base on LC networks were applied to the antennas. Figure 5b,c show S11 parameter of the proposed antenna was obtained below −60 dB each mode.

The resonant frequencies have a current distribution such that a uniform |B$_1$| field can be produced, the magnitude and phase of the current density J for the 300 MHz and 1.06 GHz is show in Figure 6, a uniform current density can be seen at the middle patch plane, thus the reason for selecting a non-uniform meander pattern.

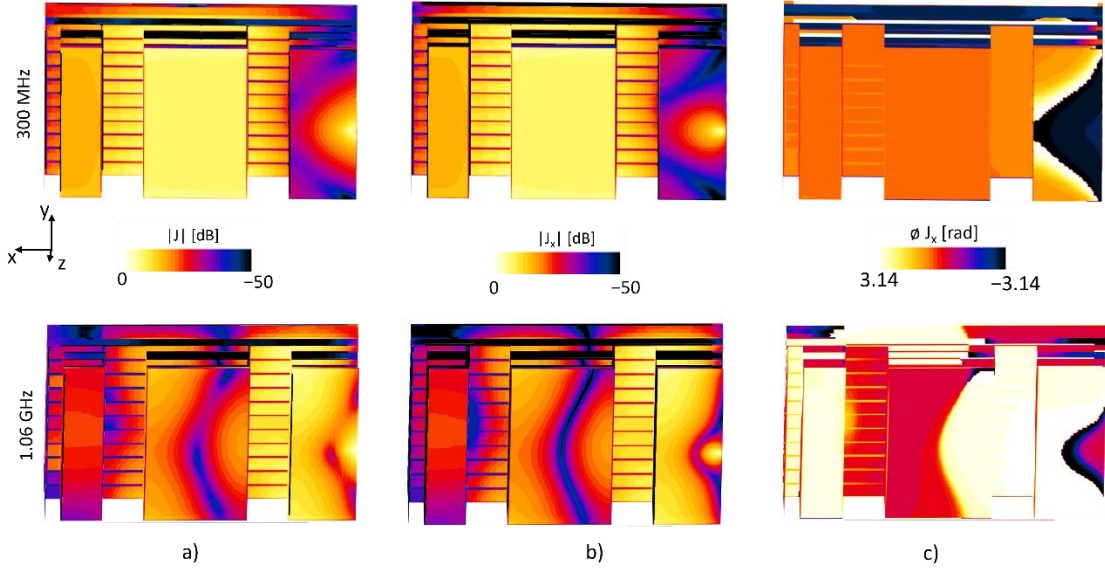

**Figure 6.** The current density distribution J of the proposed patch antenna. Showing in the top of the figure the 300 MHz and bottom the 1.06 GHz, for (**a**) the vector field of the current density, (**b**) the x component of current density, J$_x$ and (**c**) the phase of J$_x$.

### 3.1. Phantom Simulations

The $|B_1|$-field in the phantom at 300 MHz and the $|E|$-field at 1.06 GHz are shown in Figure 7, while Figure 7c shows the temperature maps and in Figure 7c the temperature difference between the normal tissue and the target cancer tissue.

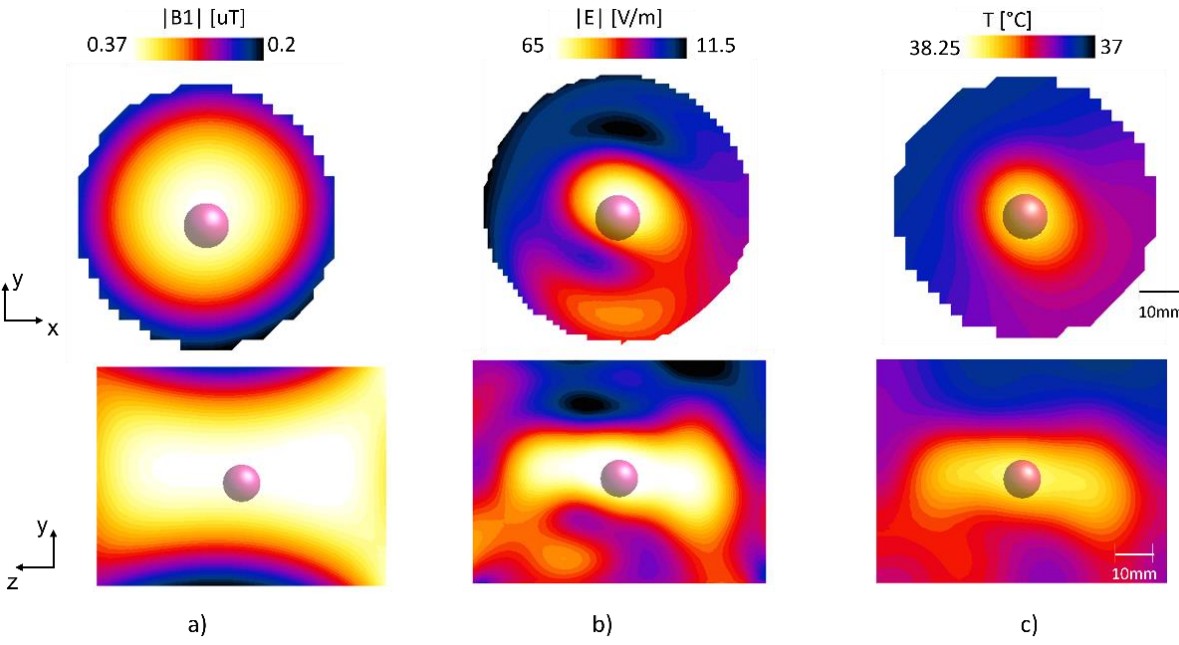

**Figure 7.** The computed (**a**) $|B|$-field, (**b**) $|E|$-field and (**c**) temperature map for the cylindrical phantom.

The $|B_1|$-field mean value for the whole phantom at the center slice was 0.3 μT, with a standard deviation of 0.048 μT, showing a highly uniform field. The E field had a peak value of 71 V/m at the location of the cancer tissue, whereas the temperature after 200 s at the cancer tissue was 38.3 °C. The mean value of the background tissue was maintained at 37.34 °C, whereas the mean value of the cancer-mimicking tissue was 38.1 °C.

### 3.2. Rat Simulations

Figure 8 shows the $|B1|$-field, $|E|$-field and temperature map under the same conditions ($S_{11}$) as the phantom simulation.

The proposed patch antenna configuration can produce a uniform $|B_1|$-field in the target area; furthermore, it shows that it can also be used for RF heating applications. The $|B_1|$-field in the target area had a mean value of 0.27 μT and a standard deviation of 0.0082 μT. The electric field had a maximum value of 400 V/m, and the temperature had an average value of 37.8 °C and maximum value of 38.2 °C in the cancer tissue after 200 s. For the background tissues the average temperature was 37.35 °C with a standard deviation of 0.15 °C and maximum of 37.9 °C. The safety of the surrounding tissues must be maintained; thus, we compared the temperature rise between the cancer tissue and healthy tissue. Figure 9 shows the SAR average 10 g maps for the case of the 300 MHz and 1.06 GHz, for the z-y and x-y plane, top and bottom, respectively. From Figure 9 it can be seen that the SAR is larger in the location of the cancer-mimicking tissue. The maximum SAR value was 0.25 and 6.32 W/kg for 300 MHz and 1.06 GHz, respectively.

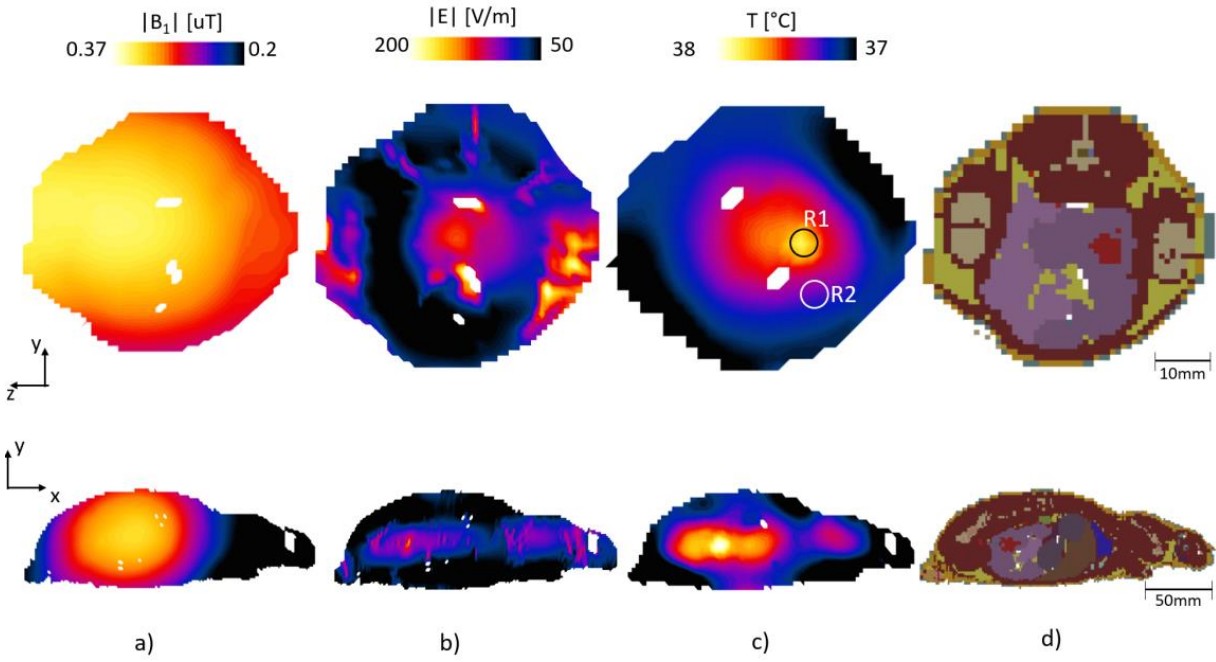

**Figure 8.** The computed (**a**) |B$_1$|-field, (**b**) |E|-field and (**c**) temperature map, for (**d**) voxel of rat model.

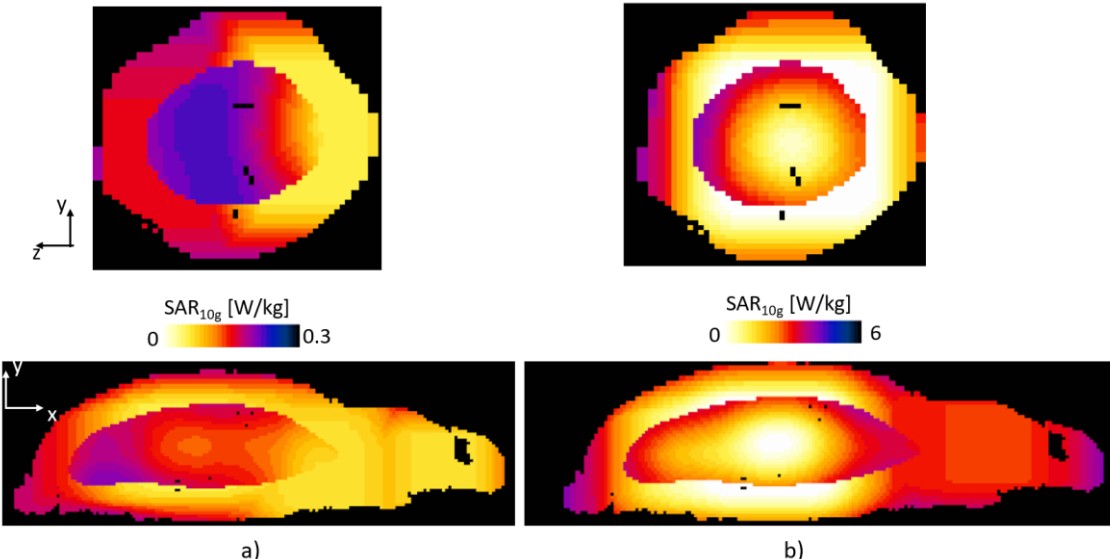

**Figure 9.** The SAR average 10 g acquired at (**a**) 300 MHz and (**b**) 1.06 GHz, for the z-y plane (**top**) and x-y (**bottom**).

Figure 10 compares the evolution of temperature in time for a point in a nearby healthy large intestine tissue (R2) and a point at the cancer tissue (R1), at the regions marked by a circle in the top of Figure 8c. In addition, we included the volume histogram to show the evolution of the temperature in the cancer tissue Figure 10b and also on the healthy tissue Figure 10c. The histogram shows the normalized volume at different temperature ranges. The bars in each bin represent the time in seconds. It can be seen that the temperature on the cancer tissue has move volume counts at the highest temperature range than in the healthy tissue.

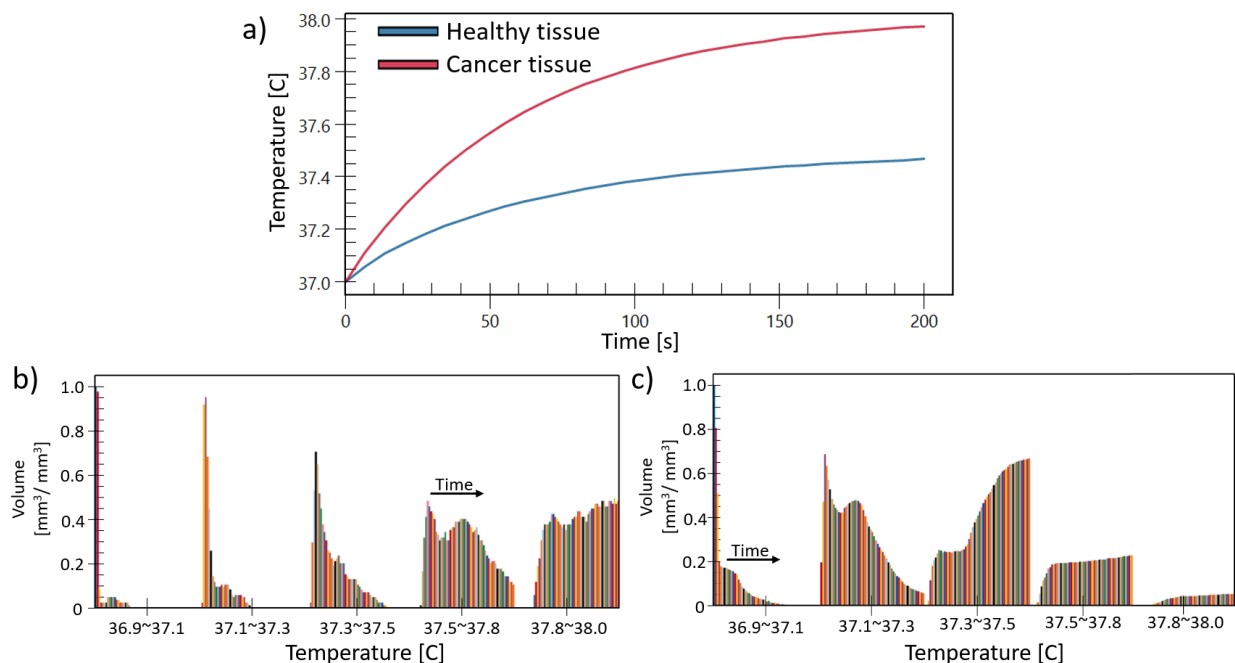

**Figure 10.** Temperature comparison as a function of time between healthy tissue (without RF heating focus) and cancer tissue (with RF heating focus) (**a**). The histogram of the temperature change on time in terms of normalized volume, for the case of (**b**) the cancer tissue, (**c**) the healthy tissue.

## 4. Conclusions

In this study, we presented a reduced-size patch antenna that can be used for MRI and RF heating. The size of the patch antennas was reduced by 61% width and 28% length using a meandering pattern in both the horizontal and vertical directions on the ground and patch plane. The designed patch antenna could produce a resonance mode at a frequency of 300 MHz corresponding to 7T MRI system. At this frequency the $|B_1|$-field is uniform for which it can be used for MR image. At a same time, the antenna had a resonance frequency at 1.06 GHz, at which the $|E|$ field is used for RF heating procedures. We verified each resonant mode and determined the one corresponding to the dominant $|E|$-field such that it can be used for the application of RF heating. We also proposed a two-patch antenna configuration capable of producing a circularly polarized and uniform $|B_1|$-field. However, this configuration can only be applied for small animal applications because the antennas are arranged along the $z$-axis and the position of the animal can be more easily controlled. The double patch antenna configuration also produced a strong and focused $|E|$-field, such that it could induce a temperature increase in a target cancer mimicking tissue. By reducing the size of the antenna, we have proposed a challenging antenna design; however, this can be achieved by using layers of copper tape attached to the dielectric material and then soldering each other. We can also create complex structures by using a 3D printer. A double tunned matching circuit can be developed for this application; however, it is outside of the scope of this work. For the case that the Hyperthermia procedure is applied in a separate time than MR imaging, an extra switch can be used to select corresponding the matching circuits. The proposed design of the patch antenna configuration can be used for combination of MRI acquisition for temperature measurement and hyperthermia with RF heating for treatments or preclinical studies with rats or mice. In the future, we will research on optimization of size adjustment for operating frequency, input power and exposure time for hyperthermia.

**Author Contributions:** Conceptualization, D.H.; methodology, D.K.; validation, D.K. and D.H.; formal analysis, D.K.; writing—original draft preparation, K.-N.K.; writing—review and editing, K.-N.K.; funding acquisition, K.-N.K. All authors have read and agreed to the published version of the manuscript.

**Funding:** This work was supported by the Institute for Information and Communications Technology Promotion (IITP) grant funded by the Korean government (MSIP) (No. 2021-0-00490, Development of precision analysis and imaging technology for biological radio waves) and supported by Gachon University under the Gachon University research fund of 2018 (GCU-2018-0672).

**Institutional Review Board Statement:** Not applicable.

**Informed Consent Statement:** Not applicable.

**Data Availability Statement:** Not applicable.

**Conflicts of Interest:** The authors declare no conflict of interest.

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
