# Peer review of "Design of a Dual-Purpose Patch Antenna for Magnetic Resonance Imaging and Induced RF Heating for Small Animal Hyperthermia"

_applsci, doi:10.3390/app11167290_

Round 1

Reviewer 1 Report

General comment:

From the lines 8-15, the paper seems to be focused on a strategy for reducing the size of patch antennas for MRI. However, the paper deals with the design of a patch antenna for performing, contemporary, 7T MRI during microwave Hyperthermia at 1.06 GHz.

The work can be interesting, however several flaws in the Introduction, Methods and Reulsts section hamper the understanding of the value and quality, as well as teh potential, of this paper. 

The author performed numeric test. Please mention that, it seems that the antenna was investigated in vivo. 

Specific comments throughout the paper:

Abstract

Line 21: Please provide the percentual standard deviation. The value, reported as it is in the abstract, has poor meaning to the general public and may appear misleading. Please revise.

  1. Introduction

Line 29: "used for application in cancer therapy" - Please, explicitely state that MRI is now the most promising tool for performing non-invasive thermometry during the hyperthermia treatment. 
This is required for ensuring a high quality (superficial hypethermia) treatment, as required by quality assurance guidelines:

Trefná, Hana Dobšíček, et al. "Quality assurance guidelines for superficial hyperthermia clinical trials: I. Clinical requirements." International Journal of Hyperthermia 33.4 (2017): 471-482.

Trefná, Hana Dobšíček, et al. "Quality assurance guidelines for superficial hyperthermia clinical trials." Strahlentherapie und Onkologie 193.5 (2017): 351-366.

Furthermore, I must report that the references [11], [12] are quite dated. You must update and improve this introduction by reading:

Curto, Sergio, et al. "Quantitative, multi-institutional evaluation of MR thermometry accuracy for deep-pelvic MR-hyperthermia systems operating in multi-vendor MR-systems using a new anthropomorphic phantom." Cancers 11.11 (2019): 1709.

Sumser, Kemal, et al. "The potential of adjusting water bolus liquid properties for economic and precise MR thermometry guided radiofrequency hyperthermia." Sensors 20.10 (2020): 2946.

Adibzadeh, Fatemeh, et al. "Systematic review of pre-clinical and clinical devices for magnetic resonance-guided radiofrequency hyperthermia." International Journal of Hyperthermia 37.1 (2020): 15-27.

Feddersen, Theresa V., et al. "Clinical performance and future potential of magnetic resonance thermometry in hyperthermia." Cancers 13.1 (2021): 31.

Lines 33:The rationale and biological effects of hyperthermia can be expanded. The authors must consider that the they are also presenting a radiator for performing hyperthermia treatment. I suggest the authors to read:

Kok, H. Petra, et al. "Heating technology for malignant tumors: A review." International Journal of Hyperthermia 37.1 (2020): 711-741.

Line 38: Missing reference for the Pennes' Equation. I suggest:

Lodi, M. B., et al. "Towards the robust and effective design of hyperthermic devices: Case study of abdominal rhabdomyosarcoma with 3d perfusion." IEEE Journal of Electromagnetics, RF and Microwaves in Medicine and Biology (2020).

Please, use the syle of this work for presenting the equation. The units for k, Q and other symbols are missing. 

Lines 39, 45: No indent after the equation. Please read the instruction for authors:

https://www.mdpi.com/journal/applsci/instructions

Line 44, Eq. (2): Does not use the cross symbol (x) for the product. Maybe dot product. Moreover, the E vector should be in norm for the SAR calculation. Correct Eq. (2). See Lodi et al. 

Line 45: Why ?? for the tissue density?

Lines 48-50: Please refer to:

Paulides, M. M., et al. "A printed Yagi–Uda antenna for application in magnetic resonance thermometry guided microwave hyperthermia applicators." Physics in Medicine & Biology 62.5 (2017): 1831.

Therein the authors can find the techical requirements for printed antennas to be used for MRI and hyperthermia.

Eq.(3)-(5) are well known, quasi scholastic and classical, for patch antennas. Reporting them explictely in the Introduction, together with the results from Fig. 1, is strange and not appropriate. 

Line 53: please cite some reference for the equations (if you decide to modify the manuscript by revising the introduction and then provide a new section for the pacth-hyperthermia issues). I suggest:

Balanis, Constantine A. Antenna theory: analysis and design. John wiley & sons, 2015.

Lines 78-79: About the methods for reducing the sizes of the patch anntennas, I must suggest the author to read:

Khan, Muhammad Umar, Mohammad Said Sharawi, and Raj Mittra. "Microstrip patch antenna miniaturisation techniques: a review." IET Microwaves, Antennas & Propagation 9.9 (2015): 913-922.

See Tab. I and Fig 11 at page 920.

The work is missing a fair comparison with techniques known in the electromagnetic and antenna engineering fiel (e.g., use of shorting pins, substrate integrated technology, use of parasitic elements, slots, magnetodielectric substrates, )

The Introduction section contains too many equations, which are typical of Methods, and therefore is poorly focused and catchy. 
In my opinion the authors are not comparing their idea with other patch antennas (used for MRI or Hyperthermia, or for both of them). A summary table would be very appreciated, please consider to use:

Paulides, M. M., et al. "A printed Yagi–Uda antenna for application in magnetic resonance thermometry guided microwave hyperthermia applicators." Physics in Medicine & Biology 62.5 (2017): 1831.

Curto, Sergio, et al. "Quantitative, multi-institutional evaluation of MR thermometry accuracy for deep-pelvic MR-hyperthermia systems operating in multi-vendor MR-systems using a new anthropomorphic phantom." Cancers 11.11 (2019): 1709.

Lodi, M. B., et al. "Towards the robust and effective design of hyperthermic devices: Case study of abdominal rhabdomyosarcoma with 3d perfusion." IEEE Journal of Electromagnetics, RF and Microwaves in Medicine and Biology (2020).

Adibzadeh, Fatemeh, et al. "Systematic review of pre-clinical and clinical devices for magnetic resonance-guided radiofrequency hyperthermia." International Journal of Hyperthermia 37.1 (2020): 15-27.

In this way, by presenting an objective comparison in terms of sizes, working frequencies, possibility to operate in MRI, you would help the readers (and the reviewers) in understanding the value, quality and innovative character of your work.

2.  Methods

Lines 90-91: The authors keep mention the modes of the antennas, however, no formula or equations are reported. Please, use:

Hazdra, Pavel, et al. "Advanced modal techniques for microstrip patch antenna analysis." 2010 Conference Proceedings ICECom, 20th International Conference on Applied Electromagnetics and Communications. IEEE, 2010.

Elsewe, Mohamed M., and Deb Chatterjee. "Modal analysis of patch slot designs in microstrip patch antennas." 2016 IEEE/ACES International Conference on Wireless Information Technology and Systems (ICWITS) and Applied Computational Electromagnetics (ACES). IEEE, 2016.

Bhattacharyya, Arka, et al. "Bandwidth Enhanced Miniaturized Patch Antenna Operating at Higher Order Dual-Mode Resonance Using Modal Analysis." IEEE Antennas and Wireless Propagation Letters (2020).

as reference.

I would like to see the current distribution (magnitude and phase) on the patch at the resonance frequencies. This is relevant to the modal analysis. 

The quality of Fig. 2 can be improved. 

Missing axes orientation in Fig. 4.

We do not have the methodological details of the simulations with the rat models.  Which software was used (commercial, in-house, Time Domain, Integral Equation..) for simulation? Which settings were used? 

I think that the authors must provide all these relevant aspects, especially the coupling between electromagnetic and thermal models. About the thermal model, the authors must clarify i) the bounary conditions at the mouse skin-air interface, ii) the absence of a water bolus (which is a strange thing for hyperthermia applications - see Sumser, Kemal, et al. "The potential of adjusting water bolus liquid properties for economic and precise MR thermometry guided radiofrequency hyperthermia." Sensors 20.10 (2020): 2946.) and iii) if the thermal parameters are assuemed constant and how the tumor perfusion was modeled (see Lodi, M. B., et al. "Towards the robust and effective design of hyperthermic devices: Case study of abdominal rhabdomyosarcoma with 3d perfusion." IEEE Journal of Electromagnetics, RF and Microwaves in Medicine and Biology (2020). ). 
The discussion of this point is mandatory. 

3. Results

Ok, only in the results you are mentioning Sim4Life. Please, provide all the methodological aspects in a clear and precise way.

Fig. 5: The antenna bandwith (evaluated at -10 dB) is null at both frequencies. This antenna is not working properly. The antenna seems to be scarcely matched. 

For what I understood, despite the strange description of the methods, the antenna is supposed to be placed in fron of the lossy, voxel rat model. Therefore, it is possible that the absence of the water bolus is responsible for this bad perfrormances. Please, see how the antenna in 

Lodi, M. B., et al. "Towards the robust and effective design of hyperthermic devices: Case study of abdominal rhabdomyosarcoma with 3d perfusion." IEEE Journal of Electromagnetics, RF and Microwaves in Medicine and Biology (2020). 

was designed considering the distance from the bolus/phantom. 

If I was wrong about the configuaration, please, pardon me. However, at this point, the authors must provide more details about how the S11 magnitude was obtained (for which of the dual patch anntennas from Fig. 4, in which coniditions).    

Lines 162-163: About the use of 5 W and 400 s for the animal system, the authors must provide a comparison with other hyperthemia treatment studies on animals (simualted and in vivo). Please support your choices.

Fig. 6: Provide the geometrical details and coordinate system.

Fig. 7: Please provide geometrical details and coordinate system. For example, you could add a (d) figure with the anatomical cross section of the voxel model, indicating the type of tissues involved. 

Fig. 8: Are the results related to the average temperature? Is a point, a minimum or maximum temperature? Please provide these details. See the work of Lodi et al. for checking the use of other hypethermia indicators, such as T50, 590. See the work of Kok et al. for CEM43 and TRISE.

A concern is that the antenna is not properly working and incereasing the tumor tissue temeprature in the range 40-44 °C (see Kok et al. 2020). The antenna cannot perform hyperthermia treatment, unless the input power is increased. I believe that the antenna is not matched.

Fig. 6 and Fig. 7: please provide the rate of decrease of the normalized B1+ field and compare it to other instrumentation used for MRI, e.g.:  

J. T. Vaughan, J. R. Griffiths, eds., RF coils for MRI, John Wiley & Sons, Frederiksberg, Denmark, 2012

J. R. Corea, et al., "Screen-printed flexible MRI receive coils," Nature Communications, vol. 7, no. 1, pp. 1-7. 2016

For MRI, also the SAR values are relevant, please provide also the 2D maps and values in target tissues for the SAR. The final temperature increase is not sufficient. Remember that you have two applications to cover, please try to use the metrics and the figure of merits trypical of each application. You can rely on the suggested readings.

After all these modificaiton, the Conlusion section will be improved accordingly. 

Additional Relevant Notes: 

1.06 GHz is not a typical working frequency for hyperthermia (see Kok et al. 2020). 915 MHz is a ISM band which can be used for the treatment. Please, provide a coherent discussion about this point. Your design may not be translated in clinical practice.

The authors are providing a meandering strategy for reducing the patch size. However, the antenna performances are scarce, they did not realized the device. I belive that a discussion on the manufacturing strategy is needed. 
If possible, I would like to see the realized antenna.

Author Response

We thank the reviewer for the thoughtful and detailed review. We have revised the manuscript as suggested by the reviewer. We have taken all the reviewer’s comments into consideration and have changed the manuscript accordingly. Specific answers to the reviewer's response area are attached as a file.

Reviewer 2 Report

This paper is suitable for publication.

Author Response

We thank the reviewer for the thoughtful and detailed review. We will supplement ourselves.

Reviewer 3 Report

The authors propose a new patch antenna design to support small animal MRI at 7T and RF hyperthermia at 1.06 GHz. While the approach from the engineering side certainly seems interesting, the presentation and the data provided to back up claims need substantial improvement. The literature cited in this work is scarce at best, which especially shows in a lack of data analysis regarding the RF hyperthermia performance of the antenna pair. I would encourage the authors to substantially rework and improve the manuscript before re-submission in order to introduce the antenna design in a more outstanding work.

Please refer to the provided document for more detailed feedback.

Author Response

(The authors gave the same response as above.)

Round 2

Reviewer 1 Report

General comment:

The author improved their work, but some issues and points should still be addressed. The list of the aspects to be improved follows. 

Specific comments throughout the paper:

Line 15 and line 18: Missing space, e.g. "electric("

Line 48: Wrong unit for the tissue density "Kg" (and not kg) instead of kg/m^3. Please fix.

Line 52: Please rephrase "In [21] demonstrates" as "In ref. [21] it is shown/demonstrate that .."

Line 54: Missing space "64MHz".

Lines 63-64: This part is a typo. It should be removed since you removed the equations for patch design.

Line 85: Reference after the stop mark. Please fix.

Eq. (3) - please be coherent with the symbols (change lx in line 104 by using the italic).

Lines 125: Please check "an FDTD", correct "Maxwell's equations"

Lines 148-149: missing reference for these values of the homogeneous phantom. How this update of the electromagnetic properties was achieved?

Line 163: This point is not clear. The authors claim that they have set a bodundary condition of the Neumann type, i.e. a convective heat transfer. Howevever, the heat flux is 0. This means that the system is isolated. Or, maybe, am I misunderstanding the fact that they are imposing that the (normal) heat flux from the surface to the air should null, so that a finite temperature value is found? Please explain. 

Line 164: The heat transfer rate is in strange unit and seems a high value. I have some problem in understanding that. The rate of energy transfer per unit time is W, you are considering also the volume in this rate (I suppose you are considering the unit volume). I can't get the K unit. 
Conventionally, the heat transfer coefficient for convective heat transfer from (human) skin to air, accounting for radiation, is 7.7 W m^-2 K^-1. This way of presenting the info is more clear and more understandable across researchers with different backgrounds.

Lines 166-167: No references for the properties of cancerous tissues. Please provide an explanation. 

Fig. 5: please increase the font size and use the same scales for sub-figures b) and c). 

Fig. 6: the caption is wrong. Please fix and be coherent with lines 186-189. 

Fig. 7 and 8: Good for the addition of the coordinate systems. Please, provide a lenght scale for understanding the level of temperature gradients in the system. I suggest you to follow the methods of microscopy image, i.e. to provide a bar to the right of the figure which can be used as a unit and reference for measuring the space.

Fig. 9 (Point 27): I am not satisfied of your reply. You are taking (lucky?) points of the two tissues, this is a very local, and poorly significnt information. I suggested more fair and conservative figure of merits which are adopted in the treatment planning and numerical simulations since they are more informative of the heat trasnfer scenario in target and non-target regions. Please, provide at least the average tumor temperature, with the relative standard deviation in the volume. 

Reviewer 3 Report

The authors have improved the quality of the manuscript by streamlining some parts of the introduction and providing more details in the methods and results section. However, significant changes still have to be made before this manuscript is ready for publication.

I have to repeat three main points of feedback already provided in the previous round:

  • Please refrain from using equations and figures in the introduction. Use a theory section instead, if you think it is necessary to repeat that much detail on the background of patch antennae
  • Please introduce more background to rat imaging at 7T (especially in the given context) and RF hyperthermia of rats to introduce a reference frame for your results. This should also include typical temperature ranges targeted for hyperthermia. Please also state (in the manuscript!) if the antennas are designed for use in a small bore animal scanner or a large bore human scanner and elaborate.
  • The authors have responded to my request to add more quantitative evaluation that SAR is not of concern for animal studies. While this may be true for the RF coil development, it is certainly not true for hyperthermia treatment planning (HTP). This section still requires thorough redesign and much more data to back up the claim of successful hyperthermia treatment. Both SAR and temperature can be used to quantify the HTP and the authors are strongly advised to read the following review as well as more recent publications to learn more about QA in hyperthermia in order to implement it for this study. I recommend:
    - Canters, R., et al., A literature survey on indicators for characterisation and optimisation of SAR distributions in deep hyperthermia, a plea for standardisation. International Journal of Hyperthermia, 2009. 25(7): p. 593-608.
    - Bellizzi, G.G., et al., Predictive value of SAR based quality indicators for head and neck hyperthermia treatment quality. International Journal of Hyperthermia, 2019. 36(1): p. 455-464.
    - Kok, H.P., et al., Treatment planning facilitates clinical decision making for hyperthermia treatments. International Journal of Hyperthermia, 2021. 38(1): p. 532-551.

Please see the attached file for the detailed comments.
